# Reproducing "Fair Selective Classification via Sufficiency"

## Reproducibility Summary

**Scope of Reproducibility**

In this reproducibility study we focus on the paper "Fair Selective Classification via Sufficiency". Our experiments focus on the following claims: 1. Sufficiency is able to mitigate disparities in precision across the entire coverage scale and in margin distributions, and will not increase these disparities compared to a baseline selective classification model in any case. 2. Using sufficiency may decrease overall accuracy in some cases, but still mitigates the disparity between groups when looking at individual classification scores. 3. The sufficiency-regularised classifier exhibits better fairness performance on traditional fairness datasets.

**Methodology**

As the authors have not made their code publicly available, all code was written from scratch, based on the instructions and pseudocode given in the original paper. Our code reconstruction contains code for training both the sufficiency model and a baseline model performing standard selective classification.

**Results**

We were not able to fully reproduce the results of the original paper in this setting. The numbers (accuracies, precisions and margin distributions) obtained in our experiments differ significantly from those reported in the original paper. Though differences between the baseline model and the sufficiency model are not as significant as in the original paper, our results do support the main claims about sufficiency being able to increase the worst-group precision and thus causing disparities between groups to decrease.

**What was easy**

The authors made the importance of implementing fair selective classification with sufficiency very clear. Moreover, the authors provided an in-depth mathematical background to sufficiency and selective classification, making their reasoning explicit. Finally, the authors presented their results in such a manner that allowed for straightforward comparison once we had trained the model.

**What was difficult**

Many technical details and model parameters were not specified in the original paper, and as no code was provided by the authors, these initially had to be determined by experimentation. Furthermore, some of the figures in the paper caused confusion about the exact implementation of the model.

**Communication with original authors**

As soon as we noticed we needed clarification on the hyperparameters, datasets and models, we contacted the authors via email. Initially we did not receive a reply, and eventually the authors were only able to answer some of our questions on the Tuesday before the deadline. While we re-implemented our model based on the newly supplied information, time was too short to fix the new issues that became apparent with the new model.

---

## 1 Introduction

Fair classification problems emerge when one wishes to ensure that underprivileged groups sharing some sensitive attribute, such as race or gender, are not disadvantaged against any other group with the same sensitive attribute (Lee et al., 2021). A variant of the fair classification problem is selective classification, where a model is allowed to abstain from making a decision. This is usually implemented via confidence thresholding. When the confidence threshold is higher, one should expect to see better performance on the remaining samples, as the system is only making decisions when it is very confident with regards to some confidence measure (Jones et al., 2020). However, it has been shown that while decreasing the coverage can increase overall performance, it can additionally magnify disparities between groups (Jones et al., 2020).

The paper that is central to this reproducibility study by Lee et al. (2021) proposes a method for enforcing fairness during selective classification, consisting of a sufficiency criterion and a regulariser based on mutual information. The authors claim that the method ensures that a classifier is fair, even if it abstains from classifying on a large number of samples. They demonstrate their method on four datasets, each consisting of a different type of data.

## 2 Scope of reproducibility

In this reproducibility study we focus on several claims. The first claim is that sufficiency is able to mitigate disparities in precision across the entire coverage scale and in margin distributions, and will not increase these disparities compared to a baseline selective classification model in any case. The second claim is that using sufficiency may decrease overall accuracy in some cases, but still mitigates the disparity between groups when looking at individual classification scores. Finally, the authors claim that sufficiency-regularised classifier exhibits better fairness performance on traditional fairness datasets.

Our study consists of two components:

- Code reconstruction: Since the author's code is not publicly available, all code was written from scratch in Python 3, using the instructions and pseudocode as described in the paper. Models, code and datasets are described in Section 3. Our code can be found on GitHub[1].

- Replication: The main part of our study is focused on reproducing the results in Lee et al. (2021), and to validate their observations and conclusions. Our replication results are presented in Section 4.

## 3 Methodology

First, an overview of the general sufficiency model is given, after which we discuss how the model was adapted to each of the datasets. This is followed by a discussion on how we evaluated our implementation, and finally we discuss the computational requirements.

### 3.1 Model descriptions

As mentioned before, the original paper uses a selective classification model to which the sufficiency criterion has been applied during training. The sufficiency criterion ensures that the predictive accuracy is the same for each group at each confidence level, that precision increases for each group when using selective classification and helps prevent disparities between groups when decreasing coverage.

For a binary target $Y$ and sensitive attribute $D$, the sufficiency criterion imposes a conditional independence between $Y$ and $D$ conditioned on the learned features $\Phi$, thus requiring:

$$P(Y = 1|\Phi(x), D = a) = P(Y = 1|\Phi(x), D = b), \forall a, b \in D.$$

An overview of the general sufficiency model is given in Figure 1. When training the model, depending on which data set is used, the data is first passed through either or both a pre-trained deep neural network and a two-layer neural

---

[1] https://github.com/MLRC2022FSCS/FSCS, accessed 04-02-22

network. The first layer serves as a feature extractor and the second one serves as a classifier. From these features, in addition to training a joint classifier, a group-specific classifier is trained for each $d \in D$. For each data point, a group-specific loss and a group-agnostic loss are computed. To obtain the first, the datapoint is assigned to the correct group-specific classifier, that is the one corresponding to the input's sensitive attribute $D = d$, while for the second the input is assigned to either of the classifiers based on the marginal distribution $P_D$. A combination of these losses is then used as a sufficiency regulariser:

$$L_R \frac{1}{n} \sum_{i=1}^{n} \left( \log q(y_i | \Phi(x_i); \theta_{d_i} - \log q(y_i | \Phi(x_i); \theta_{\widetilde{d_i}})) \right)$$

The overall loss function then becomes:

$$\min \frac{1}{n} \sum_{i=1}^{n} \left( L(T(\Phi(x_i)), y_i) + \lambda \log q(y_i | \Phi(x_i); \theta_{d_i}) - \lambda \log q(y_i | \Phi(x_i); \theta_{\widetilde{d_i}}) \right)$$

and is used to update the feature extractor and joint classifier.

By minimising the difference between the group-specific and group-agnostic loss, $\Phi(x)$ will be such that the group-specific models trained on it will decrease their individual biases and converge towards the same model. In binary

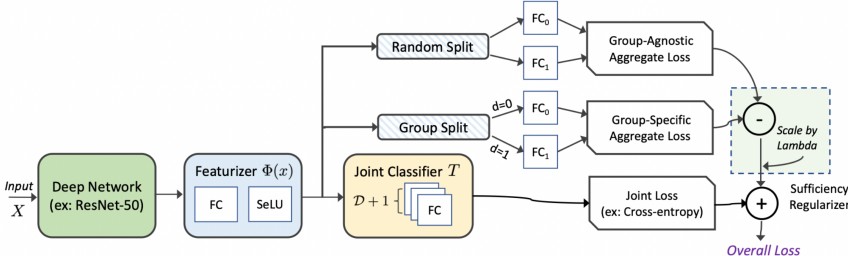

Figure 1: Overview of the sufficiency model. Obtained from the original paper by (Lee et al., 2021).

selective classification, an input $X$ is classified as belonging to a certain class when the confidence exceeds some threshold. The softmax response $s(x)$ is monotonically mapped to the confidence score $k(x)$ with the following formula, which maps $[0.5, 1]$ to $[0, \inf]$ and provides much higher resolution on the values close to 1 (Lee et al., 2021):

$$\kappa(x) = \frac{1}{2} \log \left( \frac{s(x)}{1 - s(x)} \right)$$

When $\hat{y} = y$, the margin $M(x)$ is $\kappa(x)$ and $-\kappa(x)$ otherwise. Given a threshold $\tau$, the classifier makes a correct prediction when $M(x) \geq \tau$ and an incorrect prediction when $M(x) \leq -\tau$.

## 3.2 Code reconstruction

Following the paper, our code was implemented in PyTorch[2]. This was achieved by creating the featuriser for each dataset, a joint classifier and two fully connected layers: one for the privileged and one for the unprivileged protected group. No activation layers were added to the joint classifier and group-layers, since cross-entropy loss requires logits as input. However, for evaluation of the model a softmax layer was applied to the predictions of the joint classifier as this was required for selective classification.

Three separate Adam optimisers were used: one for the featuriser, one for the joint classifier and one for both layers in the group classifiers. The loss regulariser $\lambda$ was set to 0.7 for all datasets as was stated in the paper. The learning rate was not provided in the paper, but later clarified by the authors to be 0.001 for each of the three featurisers. Moreover, there was no mention of what range was used for the confidence threshold, which determines the coverage. As such, testing starts with a threshold $\tau$ of 0 (i.e. with a coverage of 100%), and increases with some threshold step size until we reach a coverage of under 0.19. The cut-off point for coverage at 0.19 was chosen somewhat arbitrarily, as it lies a little past 0.20, which seems to be roughly the point beyond which neither the accuracy, nor the precision change much at all. This is in line with both our observations, as well as the data presented by Lee et al. (2021).

---

[2]`https://pytorch.org/docs/stable/index.html`, accessed 04-02-22

### 3.3 Dataset-specific models

We ran the experiments on three of the four binary classification datasets used in the paper. For each of the datasets, we used the predetermined train/test splits if available.

#### 3.3.1 Adult dataset

The Adult[3] dataset (Kohavi et al., 1996) consists of 48.842 entries containing tabular census data, such as age, sex and education. The first step in preprocessing the dataset was removing all entries with missing values. The data was then split into 29092 training examples and 15060 test examples. Categorical variables within the data were one-hot encoded, and the continuous variables were normalised to be between 0 and 1, the latter of which was done to remove the outliers that could incorrectly skew the gradient learning of the parameters. Following the original paper, we only kept the first 50 samples for women with a high income, that is $D = 0$ and $Y = 1$, to stimulate disparities between groups. The resulting data, $X$, was used to predict the target label $Y$, which in this case is an individual's income. Classification is binary: an income of over 50k is viewed as high income and assigned label 1, and every other income was assigned label 0. Sex was designated as the sensitive attribute $D$.

For this dataset, we followed the original paper and used a two-layer neural network with a hidden layer consisting of 80 nodes. The first layer is a feature extractor using a Scaled Exponential Linear Unit (SELU) activation function, and the second layer serves as the joint classifier. The network is trained for 20 epochs.

#### 3.3.2 CelebA dataset

The CelebA[4] dataset (Liu et al., 2015) consists of 202.599 images of 10.177 different celebrities, along with a list of attributes depicted in the images. It was not clear from the original paper whether the aligned dataset or the original one was used. Moreover, it was only specified that the images were resized to 224x224, but it was not explained how this was done and whether there were any other preprocessing steps, such as normalisation. After communication with the authors it became clear that the aligned dataset was used, and that the images were to be normalised with 0.5 mean and 0.5 standard deviation. Because the Pytorch dataloader loaded in 38 extra columns that were unnecessary in our research, we manually resized the images to 224x224 with a Pytorch transformation. In order to be able to use the cross-entropy function in a later stage, all -1 values of the binary 'blond' and 'male' variables were mapped to 0. The resulting images were used as data X, the hair colour (blond or not) was used as the target variable Y and the sex variable was used as the sensitive attribute D.

To obtain features from the images, we trained a ResNet-50 model (He et al., 2016), initialised with the pre-trained ImageNet weights, for 10 epochs. The features were then extracted from the second to last layer. The last layer was removed and replaced with a layer consisting of two output nodes to form the classifier.

#### 3.3.3 Civil Comments dataset

The Civil Comments dataset[5] (Borkan et al., 2019) is a text-based dataset consisting of 1.999.514 online comments on various news articles, along with metadata about the commenter and a label indicating whether the comment displays toxicity or not. The Kaggle repository does not provide a test set with labels nor a validation set. This meant that exclusively datapoints from the training set were used in our study. Following Lee et al. (2021), we let $X$ be the comment text, $Y$ be the binary toxicity label, and $D$ be whether Christianity is mentioned. The dataset does not include mention-of-Christianity values for each data point and therefore all data points without one were dropped. A total of 235.087 comments remained. These were subsequently split into a training, validation and test set using ratios of $0.8, 0.1, 0.1$ respectively. Additionally, the targets $Y$ and mentions of Christianity $D$ were converted to binary values, where values above or equal to $0.5$ were mapped to 1 and values below $0.5$ were mapped to 0. Lastly, the comments $X$ were tokenised using the BERT tokeniser[6], with max length set to 512, truncation set to true, and padding set to the max length.

---

[3] https://archive.ics.uci.edu/ml/datasets/adult, accessed 04-02-22

[4] http://mmlab.ie.cuhk.edu.hk/projects/CelebA.html, accessed 04-02-22

[5] https://www.kaggle.com/c/jigsaw-unintended-bias-in-toxicity-classification/data, accessed 04-02-22

[6] https://huggingface.co/docs/transformers/model_doc/bert#transformers.BertTokenizer, accessed 04-02-22

To obtain features from the texts, the tokenised data was passed through a BERT model (Devlin et al., 2018) using the pretrained parameters. The exact BERT model was not specified in the original paper. Due to time constraints, the Tiny BERT model from Hugging Face[7] (Turc et al., 2019; Bhargava et al., 2021) was used, which had previously been adapted to Pytorch. Similarly to the Adult dataset, we then applied a two-layer neural network to the BERT output with 80 nodes in the hidden layer. The first layer was treated as a feature extractor and the second layer as the classifier. Following the original paper, we trained the model for 20 epochs.

### 3.4 Evaluation

To make sure our implementation is correct, we also implement a standard classification baseline where we only optimise the cross-entropy loss function. This can be observed in the lower part of Figure 1. Moreover, we plot the margin distributions of our sufficiency implementation and compare them to that of the original paper.
To measure the effectiveness of our selective classification implementation, we follow the evaluation method of the authors and plot the accuracy-coverage and precision-coverage curves, and then compute the area under the curves to summarise the performance across coverage values.

### 3.5 Computational requirements

The experiments were run using a Nvidia RTX 3090 with 24 GB VRAM at 1785 MHz. The batch sizes were not provided in the original paper, and so they were chosen based on memory constraints. As the Adult dataset consists of relatively little data, the batch size was set to 32 in order to perform enough gradient steps to fit the parameters. This resulted in a total training runtime of about 10 seconds for the baseline and 5 minutes for the sufficiency implementation across all 20 epochs. For the CelebA dataset, the largest batch size that fit in memory was 96, which resulted in a total training runtime of 1 hour and 30 minutes for the baseline and 3 hours and 30 minutes for sufficiency for 10 epochs. For the Civil Comments model, a batch size of 48 was used, resulting in a total of 30 minutes of training time for the baseline model and 1 hour and 38 minutes for the sufficiency implementation when running for 20 epochs.

## 4  Results

### 4.1 Overall accuracy-coverage graphs

Figure 2 displays the overall accuracy plotted against the coverage for different datasets and for both the baseline and the sufficiency-regularised model. From the Adult dataset graph, we can infer that accuracies are the same for both models across all coverages. For the CelebA dataset, the sufficiency model increases the accuracy for most of the coverage scale, only converging with the baseline at a coverage of around 0.25. In the Civil Comments dataset graph, the baseline model outperforms the sufficiency model across the entire coverage scale.

In the original paper, the authors claim that sufficiency may decrease accuracy in some cases. Specifically, they show that the baseline model outperforms the sufficiency model on overall accuracy for the Adult dataset. Our results do support this specific result on the CelebA dataset, though for the Adult dataset the baseline and sufficiency models perform equally. For the Civil dataset, however, it is the case that sufficiency decreases accuracy, which thus confirms the general claim that sufficiency does not necessarily improve accuracy.

### 4.2 Group-specific precision-coverage curves

Figure 3 shows the group-specific precisions across the entire coverage scale. When comparing the baseline model to the sufficiency model, Figure 3a shows that, from a coverage of below about 0.7, sufficiency leads to a smaller gap between the female and male precisions on the Adult dataset. The worst-group precision, i.e. the male precision, improves most.
For the CelebA dataset in Figure 3b, we observe both groups' precisions increasing when using sufficiency. The precisions increase equally however, causing the gap between the genders to remain the same.
As was to be expected from Figure 2c, both precisions decrease when using sufficiency on the Civil Comments dataset. However, the gap between the two groups very slightly decreases for coverages between 0.7 and 1.0.

---

[7] `https://huggingface.co/prajjwal1/bert-tiny`, accessed 04-02-22

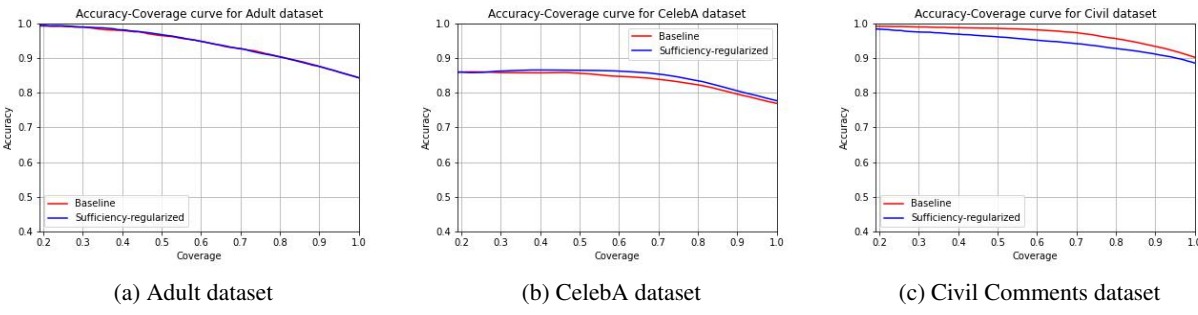

(a) Adult dataset  (b) CelebA dataset  (c) Civil Comments dataset

Figure 2: Overall accuracy-coverage graphs.

These findings are mostly in line with the findings in the original paper: while for the Adult dataset and the Civil Comments dataset the gaps between the two group do decrease when including sufficiency, and while for the CelebA dataset this is not the case, sufficiency does not increase the gap but does significantly improve accuracy. These results neither confirm nor deny the authors' claim that the sufficiency criterion introduces a method for mitigating the disparity in precision, though we do note that the differences in precision in our results are much less significant than those as reported in the original paper.

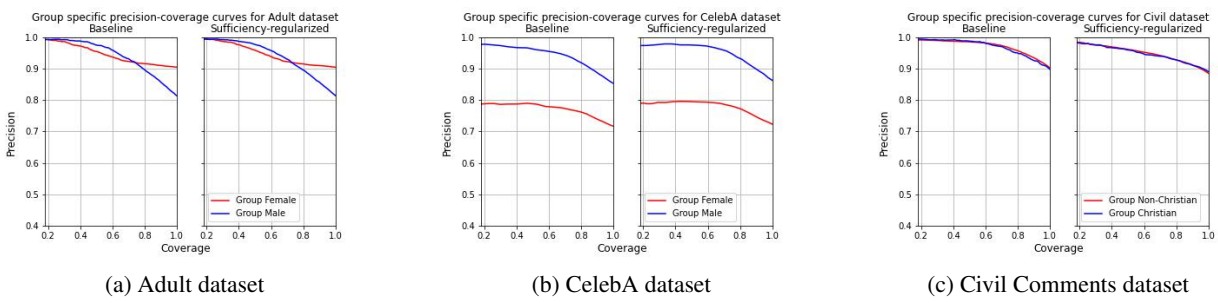

(a) Adult dataset  (b) CelebA dataset  (c) Civil Comments dataset

Figure 3: Group-specific precision-coverage graphs.

### 4.3 Margin distributions

In Figure 4, the margin distributions for both groups are displayed for each of the datasets. For the Adult dataset, the margins do not appear to be affected much by sufficiency.

Conversely, in the CelebA dataset, both margin distributions become more positively centred, causing the distributions to be more similar. Especially the male group margin shifts more towards the positive side, obtaining a smaller range in the negative region and a wider peak in the positive region. The number of samples in the female group with a negative margin has decreased. We also observe an increase in the number of outliers in the positive region.

Finally, the Civil Comments dataset shows the Non-Christian group's margin becoming more normally centred around a margin of around 1, and also shows the two groups' distributions becoming more aligned.

Our results show sufficiency mitigating and in any case not worsening disparities between the two groups, with the Adult dataset distributions staying the same and the other two datasets confirming that sufficiency causes a slight reduction of the gap between the margin distributions of different groups. This thus confirms the claim that sufficiency helps mitigate disparities in margin distributions, however, again, the differences between the models' distributions are not as clear as in the original paper.

### 4.4 Numerical evaluations

In Table 1, the areas under the accuracy curves and the areas between the precision curves are presented for each of the datasets. For the Adult dataset, the area under the accuracy curve virtually remains the same when using sufficiency, in line with Figure 2a. The area between the precision curves slightly increases. While this seems to contradict Figure 4a, note that we only observed a decrease in the precision gap for coverages of below 0.7, and the numbers in Table 1

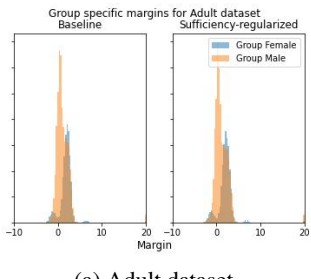 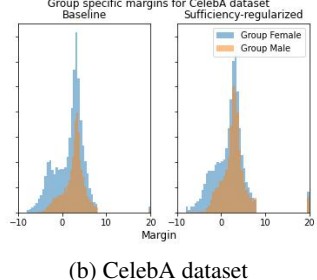 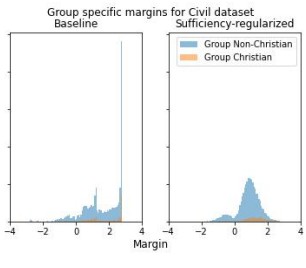

(a) Adult dataset            (b) CelebA dataset            (c) Civil Comments dataset

Figure 4: Margin distribution graphs.

211 concern the entire coverage scare.
212 For the CelebA dataset, the area under the accuracy curve increases, resulting in an increase in overall accuracy as
213 previously observed in Figure 2b. However, as already suggested by Figure 4b, the area between the precision curves
214 stays virtually the same when using the sufficiency method.
215 Once again confirming the results observed in Figure 3, when using sufficiency, the area under the Civil Comments
216 dataset's accuracy curve decreases. The area between the precision curves effectively stays the same.

217 As mentioned before, in the original paper sufficiency causes the Adult dataset accuracy to diminish, while in our results
218 both models achieve the same performance. In contrast, while for the Civil Comments dataset the original paper's
219 accuracy increases, our results show a decrease in accuracy. The CelebA results both exhibit an increase in accuracy,
220 though this increase is more prominent in the original paper.

221 The area between the precision curves significantly decreases for the Adult dataset in the original paper, which is not the
222 case for our results. The same holds for the precision curves of the the CelebA and Civil Comments dataset: although
223 less so than for the Adult dataset, the original paper's result show that disparities are decreased when using sufficiency,
224 while our results do not show any significant change.

225 The Civil Comments results show that accuracy can reduce when using sufficiency, but disparities will not increase.
226 Furthermore, although the claim about disparity in precision mitigating is not directly confirmed by our results as the
227 areas stay the same, it does show that sufficiency will not (significantly) worsen disparities in any case.

| Dataset | Method | Area under accuracy curve | Area between precision curves |
|---|---|---|---|
| Adult | Baseline | 0.931 | 0.220 |
| | **Reproduced baseline** | **0.941** | **0.004** |
| | Sufficiency | 0.887 | 0.021 |
| | **Reproduced sufficiency** | **0.942** | **0.005** |
| CelebA | Baseline | 0.852 | 0.094 |
| | **Reproduced baseline** | **0.855** | **0.141** |
| | Sufficiency | 0.975 | 0.013 |
| | **Reproduced sufficiency** | **0.863** | **0.142** |
| Civil Comments | Baseline | 0.888 | 0.026 |
| | **Reproduced baseline** | **0.973** | **0.0012** |
| | Sufficiency | 0.943 | 0.010 |
| | **Reproduced sufficiency** | **0.954** | **0.0010** |

Table 1: Numerical comparison between original paper and reproduction.

## 5 Discussion

229 To summarise, the numbers (accuracies, precisions, margin distributions etc.) obtained in our experiments differ
230 significantly from those reported in the original paper. However, although differences between the baseline model
231 and the sufficiency model are not as significant as in the original paper, our results do support the main claims about
232 sufficiency being able to increase the worst-group precision and thus causing disparities between groups to decrease.

It is worth mentioning that the Figures 4b and 4c show the largest increase in margin alignment, and these are also the datasets that either improve in overall accuracy, or decrease in disparities between groups. Moreover, the authors claimed that the sufficiency-regularised classifier exhibited better fairness performance on traditional fairness datasets. Though we were not able to reproduce their results in this study, we do believe we can validate this claim, as sufficiency is either able to decrease the disparities in precision between groups (Figures 3a and 3c), or increase the precision for both groups in an equal manner as we traverse the coverage scale, meaning that no group is penalised for the sake of improving the other group's precision.

The fact that we were not able to precisely reproduce the results from the original paper is likely due to the fact that not all technical details required to fully replicate the original paper were provided by the authors in the paper. Specifically the learning rate, selective classification threshold and optimiser algorithms had to be decided upon ourselves. While a well-informed guess of what parameters to use was made possible due to experimentation, it could well be possible that the authors' implementation differs on these fronts, and that this caused our results to differ from the ones in the paper. We also did not have time to run all the experiments we would have liked to, such as the fourth CheXpert[8] dataset or experiments beyond replication, such as applying the sufficiency method to a new dataset. This was due to the fact that we spent a large amount of time trying to improve our original results, because we wanted to make sure these were stable before attempting to generalise further.

## 5.1 Reproducibility of the paper

### 5.1.1 What was easy

The authors provided a strong and logically structured theoretical background that made the importance of implementing fair selective classification with sufficiency very clear. Moreover, the authors provided an in-depth mathematical background to sufficiency and selective classification, making their reasoning explicit. Finally, the authors provided clear explanations of the evaluation method and presented their results in such a manner that allowed for straightforward comparison once we had trained the model.

### 5.1.2 What was difficult

As mentioned previously, many crucial technical details (e.g. pretrained models and hyperparameters) required to replicate the original paper were not provided by the authors. Furthermore, we found the overview of the model shown in Figure 1 (Figure 2 in the original paper) difficult to interpret. This caused the implementation of the model to take more time than we had anticipated. The first issue was the use of "ex" in the deep network and joint loss depictions, which is generally short for "excluding". In section 4.1 of the original paper, it appears that the ResNet-50 model is modified in place, leading to the features being extracted and classified within ResNet-50 itself. This would indeed indicate 'ex' meaning 'excluding', as there is no separate featuriser in this case. However, this interpretation means that cross-entropy is excluded from the joint loss, though it is explicitly mentioned in section 4.1. This would indicate "ex" is short for 'exemplum', which is a contradiction. Moreover, the image does not make immediately clear that the fully connected layers $FC_0$ and $FC_1$ are the same for both the group-specific and the group-agnostic classifier. There was also no mention of the loss functions or activation functions used for the fully connected layers in the group-specific classifiers. Finally, it was not explicitly mentioned whether the featurisers were the same for the Adult and Civil datasets.

## 5.2 Communication with original authors

As soon as we noticed we were missing crucial information about the hyperparameters and the CelebA dataset and we needed some clarifications on the workings of the model, we contacted the authors via email. Initially we did not receive a reply, and so we sent a follow-up email. We received an answer from the authors that they needed more time to verify the information we asked for and were currently working towards a deadline themselves and we eventually received an email on 01-02-2022. In this email, the authors were only able to answer some of our questions. While we re-implemented our model based on the newly supplied information, time was too short to fix the new issues that became apparent with the new model.

---

[8] https://stanfordmlgroup.github.io/competitions/chexpert/, accessed 04-02-22

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
