# OpenReview forum: "Reproducing "Fair Selective Classification via Sufficiency""
_ML_Reproducibility_Challenge/2021/Fall — RC2021_

### Official Review · Reviewer_cAtU · 2022-03-01
**Reconstruction of code that is not publicly available and good comparison against results in original paper**

**Rating:** 7
**Confidence:** 3

**Review:**

This paper reconstructs the code that is used in original paper and provides access for future use. They perform the experiments done in the original paper. They provide the nice summary of data processing used in the experiments.

It is slightly surprising and concerning that the curves for adult dataset don't match in both papers especially how in precision-coverage plots in original paper, precision is almost 1 for most of the coverage where as in this paper it is not the case.

In the experiments authors found that sufficiency is either able to decrease the disparities in group precisions or increase the precision for both groups but not in the same terms presented in the original paper.

But I think it is a good attempt at reproducing the results and could have benefitted from exact learning rate and algorithms used in the original paper. But their code helps as a good starting point for someone exploring this in future.

Even though this is not the original paper, it would be nice to expand on details of selective classification and regularizer based on mutual information.

Equation under line 77 is missing a bracket.

---

### Official Review · Reviewer_S82V · 2022-04-01
**Good report without code from original authors**

**Rating:** 7
**Confidence:** 4

**Review:**

The report reproduces the result of the ICML 2021 paper titled "Fair Selective Classification via Sufficiency".

Scope of reproducibility: The authors focus on validating the qualitative claims made in the original paper and confirming their claims and observations.

Code: The authors have developed the code from scratch due to the unavailability of code from the original authors.

Communication with original authors: The authors tried to connect with the original authors to clarify details but received a response late.

Discussion on results and Recommendations for reproducibility: The authors provide a detailed discussion on the results they were able to reproduce and also compared it against a baseline model that uses cross-entropy loss. They made a complete effort to verify the claims of the paper and drew insights into the validity of the results.

Overall organization and clarity: The paper is very well written and comprehensive.

---

### Meta-Review · Area_Chair_1Unb · 2022-04-09

**Recommendation:** Accept
**Confidence:** 4

**Metareview:**

Reviewers praised this work, especially that it reimplements code that wasn't released by the original paper.

---

### Decision · Program_Chairs · 2022-04-09

**Decision:**

Accept

**Comment:**

Following the recommendation of reviewers and meta-reviewer, the paper is accepted for ML Reproducibility Challenge 2021, and will be published in the upcoming special edition of ReScience Journal.